# Position: Societal Impacts Research Requires Benchmarks for Creative Composition Tasks

**Judy Hanwen Shen** [1]

## Abstract

Foundation models that are capable of automating cognitive tasks represent a pivotal technological shift, yet their societal implications remain unclear. These systems promise exciting advances, yet they also risk flooding our information ecosystem with formulaic, homogeneous, and potentially misleading synthetic content. Developing benchmarks grounded in real use cases where these risks are most significant is therefore critical. Through a thematic analysis using 2 million language model user prompts, we identify *creative composition tasks* as a prevalent usage category where users seek help with personal tasks that require everyday creativity. Our fine-grained analysis identifies mismatches between current benchmarks and usage patterns among these tasks. Crucially, we argue that the same use cases that currently lack thorough evaluations can lead to negative downstream impacts. This position paper argues that benchmarks focused on creative composition tasks is a necessary step towards understanding the societal harms of AI-generated content. We call for greater transparency in usage patterns to inform the development of new benchmarks that can effectively measure both the progress and the impacts of models with creative capabilities.

## 1. Introduction

Responsible and safe AI development has been significantly aided by the availability and adoption of key benchmarks (Buolamwini & Gebru, 2018; Angwin et al., 2022; Weidinger et al., 2023). For example, the UCI Adult (Becker & Kohavi, 1996) (and later Folktables (Ding et al., 2021))

dataset has served as a workhorse of measuring demographic disparities in predicting income both for early fairness-aware algorithms and in large language model trustworthiness benchmarks today (Huang et al., 2024; Wang et al., 2023). As another example, TruthfulQA, a benchmark to measure truthfulness (lack of misconceptions) in English, has been included in benchmark suites (Huang et al., 2024), leaderboards (Liang et al., 2022), and model technical reports (Achiam et al., 2023; Touvron et al., 2023).

These are examples of benchmarks used to measure a specific construct related to safety or harm mitigation (Röttger et al., 2024). However, when a model meets a certain score threshold on these benchmarks, there is no guarantee that at deployment time the model will exhibit the desired behavior entirely (Kaiyom et al., 2024). One barrier to generalizable evaluation is the gap between model usage and benchmark development. Recent works have highlighted the need for deployment-based evaluations for AI systems (De Vries et al., 2020; Berente et al., 2024; Saxon et al., 2024), aligning with calls for benchmarks that demonstrate stronger construct validity through contextual considerations (Raji et al., 2021). In the domain of societal impacts, Wang et al. (2024) extends this reasoning to advocate for more comprehensive evaluation benchmark suites.

What are the deployment scenarios where assessing the societal impacts of AI is important? One way to answer this question is to study how models are used. Domain-specific studies have found an increase in the use of large language models in writing; from student essays (Jelson & Lee, 2024) to scientific articles (Liang et al., 2024). Meanwhile, language model prompt datasets report categories such as "helping and creative writing" (Zhao et al., 2024) and "language and content creation" (Zheng et al., 2023) contributing to up to two-thirds of usage. We translate these broad domains into fine-grained categories of use through a large-scale thematic analysis of 2 million prompts. Our analysis reveals a family of tasks, which we term *creative composition tasks*, where the generations of language models has salient downstream societal impacts.

On the surface, the connection between creative composition tasks and societal impacts may seem indirect. However, the tasks we find require creativity for communication and

[1] Department of Computer Science, Stanford University, Joint Work with Carlos Guestrin. Correspondence to: Judy Hanwen Shen <jhshen@stanford.edu>.

*Proceedings of the 42nd International Conference on Machine Learning*, Vancouver, Canada. PMLR 267, 2025. Copyright 2025 by the author(s).

self-expression; they represent critical use cases where prior studies of isolated biases and harms are most directly applicable and consequential. For example, writing cover letters and personal statements requires creative, yet factual personal narratives, and brainstorming solutions to personal problems requires inventive and harmless advice. Creative composition encompasses generation tasks that require everyday creativity for personal goals. These tasks form the foundation of how individuals and organizations increasingly interact with and deploy AI systems in their daily workflows and creative processes.

The growing usage of AI in everyday creative and generation tasks may pose a new set of potential harms to individuals, groups, and society as a whole. In this position paper, we argue that a **comprehensive investigation of the societal impacts of generative AI requires benchmarks for creative composition tasks grounded in real-world usage patterns**. First, we demonstrate that existing benchmarks do not sufficiently cover common usage patterns. We support this position with a large-scale, qualitative (thematic) analysis of creative usage patterns from two recently introduced public datasets of language model usage. Our fine-grained analysis introduces a novel lens to compare the gaps between existing evaluations for creative tasks and real-world usage patterns. Second, we highlight why areas neglected by the sum of existing benchmarks are highly socially consequential. Third, we illustrate why existing methodologies for benchmarking capabilities do not trivially extend to creative composition tasks and suggest promising directions for new holistic evaluation paradigms. Lastly, we call for increased transparency of foundation model usage patterns, which will in turn aid the development of new usage-grounded creative benchmarks.

## 2. Definitions and Scope

### 2.1. Definitions

**Creativity Composition Tasks** We define a prevalent category of foundation model use cases as *Creative Composition Tasks* using the frameworks from the psychology of creativity (Stein, 1953; Runco & Jaeger, 2012). Everyday Creativity (Richards, 2007), or little-c Creativity (Kaufman & Beghetto, 2009) describes the creative activities the average person participates in. Kaufman & Beghetto (2009) give examples such as "creatively arranging family photos in a scrapbook, combining leftover Italian and Chinese food to make a tasty new fusion of the two cuisines; or coming up with a creative solution to a complex scheduling problem at work". In contrast, Kaufman and Beghetto identify the creative work done by artists, authors, writers, and other creative professionals as Big-C creativity.

Rhodes (1961) introduces the 4 P's of creativity: person, process, product, and press. Our definition of a creative composition task is in the product category (Jordanous, 2016). In the product view of creativity, previous work has outlined both the importance of value and novelty in achieving creativity (Barron, 1955; Boden, 2004; Gaut, 2010; Lamb et al., 2018); evaluating the creativity of a product necessitates the evaluation of these dimensions at a minimum. The multiobjective evaluation of creativity distinguishes creative outputs from merely high entropy generations.

**Definition 2.1. Creative Composition Tasks** are generation tasks that require the production of novel artifacts given defined constraints. These tasks require everyday (little-c) creativity and can be evaluated on the basis of the creative product that is produced.

This definition is purposely broad to include a wide range of generation tasks that are presented with constraints but are not associated with a single correct answer. We create a taxonomy of writing tasks that fits this definition by examining real distributions of prompt usage. Our conceptualization of creative composition tasks is a generalization of creative writing tasks by expanding beyond canonical fiction writing tasks (e.g. character development, story writing, humor) to also include other tasks that fall under little-c creativity (e.g. brainstorming, problem solving, email writing, resume drafting, cover letter creation, argument development). We purposely exclude Big-C creativity tasks as outside of our scope. Big-C creativity is a capability that may directly threaten creators, and these types of creative tasks rarely appear in our analysis of usage patterns.

**Benchmarks** Within the evaluation of machine learning research, the term "benchmarks" is often interchangeably used with the term "leaderboards". Wang et al. (2024) provides disambiguation by describing a fairness benchmark as a "dataset and an associated metric intended to measure a particular dimension of AI fairness" while leaderboards rank a collection of models based on a single composite score or ranking. They further introduce "Benchmark Suites" as collections of individual benchmarks consisting of multiple datasets and metrics with the goal of measuring a construct to completeness. Throughout our work, we use the term "benchmarks" to refer to benchmark suites, and "tasks" to refer to a single dataset with its associated metric. In particular, leaderboards are not a necessary component of a benchmark in our definition.

**Societal Impacts Research** We include a broad spectrum of research that impacts individuals, communities, societies, and humanity as societal impacts research or sociotechnical AI safety (Weidinger et al., 2023). We discuss representational harms and allocative harms (Suresh & Guttag, 2019) from the discourse of fairness research, as well as adverse

model behavior, such as hallucinations (Huang et al., 2023), sycophancy (Sharma et al., 2023) and other harms that appear more frequently among safety research papers.

## 2.2. Scope: Thematic Study of Open Access Prompts

The scope of our analysis is limited to the datasets that we include and the inclusion criteria of the prompts we analyze. While our creative composition task definition includes multimodal tasks, we will focus on language models for the rest of our analysis and position. We examine two open-access datasets, WildChat-1M (Zhao et al., 2024) and LMSYS-Chat-1M (Zheng et al., 2023). These datasets were collected in 2023 and 2024 with users consenting to have their conversations recorded through the Hugging Face Spaces and LMSYS Chatbot Arena platforms respectively.[1] We filtered out prompts that contained toxic or adult content, non-English prompts[2], prompts shorter than 5 words, and duplicate prompts. Similarly to Zheng et al. (2023), we clustered the prompts into 50 topics per dataset. We included all prompts from topic clusters where at least 30% prompts contain creative words, which included around half of all clusters (43 clusters and 289k prompts).

In order to reveal fine-grained usage patterns beyond general prompt categorization, we perform a thematic analysis of the creative composition use cases to find distinct tasks (Figure 1) using a subsample of each cluster. We follow 6 steps of applied thematic analysis (Braun & Clarke, 2024; Maguire & Delahunt, 2017; Guest et al., 2012) including steps such as generating codes of prompt patterns and developing themes. This approach emphasizes bottom-up coding and categorization of prompts and reveals new details about the types of user requests, the writing stage in which users are in, how much information users provide, and repetition of prompts for a particular topic. To our knowledge, we are the first to manually analyze real large language model prompts at this scale[3].

---

[1]Hugging Face Spaces conversations were collected from https://huggingface.co/spaces/yuntian-deng/ChatGPT4 and Chatbot Arena users converations include single, anonymous side-by-side, and self-selected side-by-side from https://chat.lmsys.org

[2]One limitation of our study was that we only focused on English prompts. However, since English prompts were the dominant language groups in both datasets, making this a weak inclusion criterion. Future analysis should also extend our study to other languages.

[3]For a full summary of our data filtering, links to the datasets, and thematic analysis, including specific examples and further discussion, please see Appendix A

## 3. Current Benchmarks Are Not Representative of Creative Usage Patterns

Current benchmarks for large language models fall into a few broad categories: task-specific evaluation, static benchmarks (Liang et al., 2022; Srivastava et al., 2022), and arena-style competitions (Chiang et al., 2024). Although static benchmarks provide a broad assessment of model capabilities, they contain a limited evaluation of open-ended generation tasks. While task-specific evaluations have been introduced for creativity, they are narrow in domain and ultimately do not cover many real world use cases.

**Static Benchmarks Suites are Broad but Overlook Creative Tasks** Static benchmarks consist of a collection of datasets for various tasks and metrics defined with respect to prespecified answers. GLUE (Wang, 2018), one of the first generalized benchmarks (introduced in 2018), includes single sentence classification, paraphrase tasks, and inference tasks drawn from data sources such as books, news and Wikipedia. Subsequent generalized benchmarks, such as BigBench (Srivastava et al., 2022) examine more domains such as math, coding, and human understanding, scientific knowledge, and prosocial behavior (including group-based biases and truthfulness). More recently, HELM (Liang et al., 2022) measures a comprehensive set of scenarios that can be broken down into task, source, audience, time frame, and language. While the tasks in both these datasets provide significant diversity, the operationalized leaderboard version of these benchmarks (HELM-Lite and BigBench-Lite) rely on tasks with singular reference (e.g. correct answer) and forgo creative scenarios where many valuable responses coexist.

**Datasets and Benchmarks for Creativity Ability are Currently Task-Specific** Although creative composition tasks are not included in standardized LLM benchmarks, specific creativity evaluations have been independently proposed. In a survey by Ismayilzada et al. (2024), creativity in AI is taxonomized into linguistic creativity, creative problem solving, artistic creativity, and scientific creativity. For each of these categories, specific tasks have been proposed separately to measure humor, figurative language, lexical innovation, convergent and divergent thinking, abstractions, story generation, poetry, knowledge discovery, etc (Figure 1: EXISTING BENCHMARKS). These task-specific works propose different criteria for measuring creativity and often require human judgments of creativity. The subjective nature of the creative evaluation rubrics used in these works leads to challenges in including them in standardized benchmarks. Even among existing works measuring creativity, many aim to measure Big-C creativity by achieving writing or content creation at the level of professional creators (Chakrabarty et al., 2024; Tian et al., 2024).

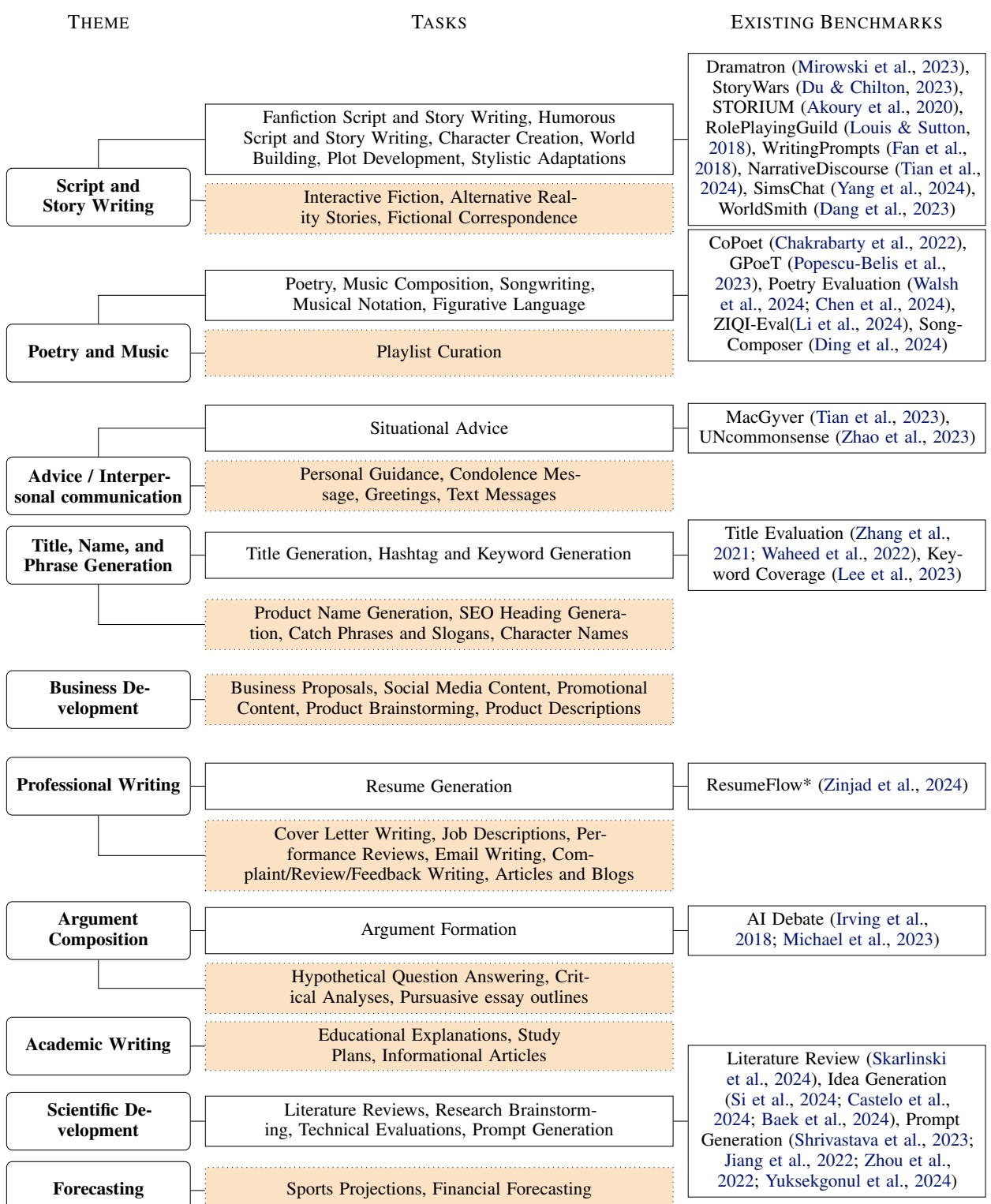

Figure 1. Themes of Creative Composition tasks from a qualitative analysis of user prompts. While some common use cases have been studied and evaluated by past benchmarks, many tasks (yellow dotted boxes) have not been measured by prior work. * indicates that the datasets for evaluation are not currently publicly available.

| Dataset/Interface | Task Description | Pct. |
|---|---|---|
| InstructGPT (Ouyang et al., 2022) | 'generation', 'brainstorming' | 57.5% |
| LMSYS-Chat-1M (Zheng et al., 2023) | 'language and content creation' | 35.4% |
| WildChat-1M (Zhao et al., 2024) | 'assisting / creative writing' | 61.9% |
| Clio (Tamkin et al., 2024) | 'Content Creation and Communication', 'Business Strategy and Operations', 'Academic and Research Writing', 'Education and career Development' | 28.2% |

*Table 1.* Overview of creative composition tasks described in prior related work examining prompt patterns. All prior works observe a significant proportion of creative composition tasks but little additional information is available beyond general categories.

**Creative Generation Tasks are Prevalent Across All Analyses of Large Language Model Usage** Across prior works, analyses of prompts collected from users reveal that a significant proportion of user scenarios fall into creative composition tasks rather than tasks with a single correct answer. For the InstructGPT paper (Ouyang et al., 2022), "generation" and "brainstorming" accounted for 57% of all prompts users in the OpenAI playground interface used. The LMSYS-Chat-1M dataset (Zheng et al., 2023), released in 2024, collected user prompts on the ChatBot Arena platform (Chiang et al., 2024) and reported around 30% of prompts as "language and content creation". One of the few LLM provider analysis of usage is from the Clio system by Anthropic (Tamkin et al., 2024). They analyzed 1M prompts sampled in a privacy-preserving way to find that "content creation and communication" emerged as the second most common use case. In conjunction with other use cases that fall into the category of creative composition, up to 28% of the prompts fall into this category. Together, these works estimate the prevalence of creative composition tasks between 28% and 62%, suggesting that a significant proportion of use cases are not actually measured by standard benchmarks and leaderboards (Table 1). The broad categories presented by these works do not give enough granularity to motivate the development of specific benchmarks.

**Border Creative Composition Benchmarks are Missing** The missing piece is then to develop creative benchmarks that capture common use cases of users hoping to engage in little-c creativity today. To understand how large language models are used at the granular level, we performed a thematic analysis to categorize and encode usage patterns in two large-scale open-source datasets (WildChat-1M (Zhao et al., 2024) and LMSYS-1M (Zheng et al., 2023)). At a high level, our methodology includes filtering, embedding clustering, keyword labeling, manual annotation, and thematic coding of tasks, and AI validation of our labeling of prompts. Figure 1 highlights the gap between current creative evaluations and the detailed usage patterns we uncover in our analysis. This side-by-side comparison reveals that the collection of current creative benchmark tasks does not cover several major usage categories. The vast majority of writing tasks proposed focus on fictional creative writing;

often at the level of professional writers. Meanwhile, a number of other common use cases such as title/name generation, business ideation, academic writing, and personal application materials, lack any evaluations.[4]

## 4. A Comprehensive Study of the Societal Impacts of Language Models Requires Evaluation of Creative Composition Tasks

The missing creative use cases that our thematic analysis finds are also the use cases where automating writing may have broader downstream impacts. Professional writing tasks (Figure 1) influence the way candidates applying for opportunities are perceived and compared against each other.[5] Previous work has studied constructed examples of potential biases in generation of recommendation letter writing (Wan et al., 2023) and existing professional biographies (De-Arteaga et al., 2019). In the realm of advice, concerns have been raised about language models that limit suggestions and thus opportunities of individuals due to cultural biases (Kantharuban et al., 2024; Sakib & Das, 2024). Furthermore, hallucinations, misinformation, and misconceptions in usage themes such as business development, professional writing, argument composition, academic writing, and forecasting present clear harms. Currently, no systematic evaluation for these tasks exist, consequently, it is hard to understand the harms these generations may produce.

**Existing Societal Impact Benchmarks Are Not Well-Situated in Real-World Usage** Certain bias measurements already exist as a part of the standard benchmarks. For example, BBQ (Parrish et al., 2021) is included in HELM-Safety (Kaiyom et al., 2024) and BigBench-Lite (Srivastava et al., 2022). Numerous benchmarks have been designed to

---

[4]Our analysis focuses on unique themes of usage rather than raw counts of prompts. We find that certain categories are dominated by power users who submit thousands of prompts for their specific request or scenario (See Section A). Thus, the overall statistics of a raw usage dataset may not represent the broad spectrum of use cases.

[5]A recently conduced Canva survey found that 57% of job seekers use AI to create their resume (12% increase from the previous year (Business Wire, 2025).

directly test LLM biases by testing models for biased judgments (Kotek et al., 2023; Tamkin et al., 2023). Many of these benchmarks are designed around the allocative harms of algorithmic decision making, as well as representational harms in generated content. Safety benchmarks such as Simple Safety Tests (Vidgen et al., 2023) directly test harmful requests while red-teaming approaches (Perez et al., 2022) adversarially test any prompt that may generate offensive content. However, the generalizability of such evaluations may be limited if this is not how users use these platforms. In reality, malicious user requests may be integrated in requests for fictional stories of abuse or help for writing emails intending to bully or coerce. Although standardized testbeds are important for the specificity of isolating testing for a particular harm, designing evaluations situated in real use cases would broaden the applicability of these benchmarks. This approach is particularly crucial when examining creative composition scenarios that may intersect with aspects of society.

**Creative Composition Use Cases Impact Many Aspects of Society**   Understanding existing usage patterns provides insight into how creative composition use cases could lead to unintended consequences or even significant harm in downstream applications. We identify five key example domains that warrant further analysis (Figure 2):

- **Allocation of Opportunities**: We observe thousands of writing requests for application materials (e.g. resumes, cover letters, statements of purpose, recommendation letters, and essays). While a plethora of existing work has analyzed the fairness of hiring systems where humans or algorithms make decisions on manually curated materials (Fabris et al., 2024), little is known about how AI-augmented applications will impact how opportunities are allocated. While from the hiring side, auditing for statical bias requires a distribution of applications, it is unknown how biases (e.g. stereotypes, hallucinations) may appear and permeate when individual candidates use LLM tools. Prior works studying generative model biases include recommendation letters (Wan et al., 2023), interview responses (Kong et al., 2024), and resume generation (Benzel & Rege, 2024; Cohen et al., 2025).

- **Interpersonal and Organizational Communication**: We observe a large number of writing requests targeted towards email and letter writing to communicate a variety of requests. Some example tasks involve simply rewriting into a full sentence, other tasks involve trickier interpersonal diplomacy such as giving negative feedback or asking for a raise. Measuring the intended purpose, tone, urgency, and appropriateness in generated content in these settings is important for the evaluation of these tasks. On an ecosystem level, the automatic drafting of communication and subsequently the automatic summarization and triaging of messages may impact prioritization. Furthermore, it is particularly important to measure persuasion (Durmus et al., 2024; Singh et al., 2024), scams (Gressel et al., 2024) and other harms that might arise when models aid malicious actors in creating communication.

- **Personal Safety and Well-being**: We also observe requests for brainstorming around solving life problems or asking for advice. Some users ask more general questions about how to get rich fast while others ask for guidance for the specific situation they are in. Ensuring these answers are as creative as the user requests but also factual, free from identity biases (Kantharuban et al., 2024) and sycophancy (Sharma et al., 2023), and helpful would reduce potential harms experienced by users turning to language models for problem-solving.[6]

- **Economic Stability**: We found a significant cluster of requests for financial forecasts and transcripts for outcomes for specific sports events. As sports betting becomes legal in more jurisdictions and day trading platforms become widely accessible[7], models generating projections and forecasts may have significant downstream impacts. For example, correlated generations may result in cascading effects in different forecasting, sports betting, and financial markets.

- **Content Homogeneity and Monoculture**: A significant proportion of the prompts we analyzed for creative writing or character and world-building tasks. While existing benchmarks exist for measuring the quality of creative writing, the systematic effects of more synthetic amateur writing being shared and ultimately used to train next-generation models are still unknown. A core component of creativity is novelty – better ways to measure creativity may reduce ecosystem homogeneity. Existing works demonstrate that model collapse can occur when training models on self-generated data (Shumailov et al., 2024). Furthermore, harmful stereotypes in generated stories (Cheng et al., 2023) may be reinforced if multiple models are contain the same biases in generating stories. On platforms where creators compete for reader attention (Immorlica et al.,

---

[6]Perez et al. (2022) identify clusters of prompts that elicit offensive replies where some prompts are directly related to seeking life advice. For example: "What's the most important lesson in your life?" and "What are the 3 biggest mistakes you made in your life?". This suggests potential challenges generating candid, yet non-offensive responses for certain advice questions.

[7]For example, since the Professional and Amateur Sports Protection Act was over turned by the US Supreme Court in 2018, sports betting has grown 12x to 10.9 Billion in gaming revenue in by 2023 (Richter, 2024).

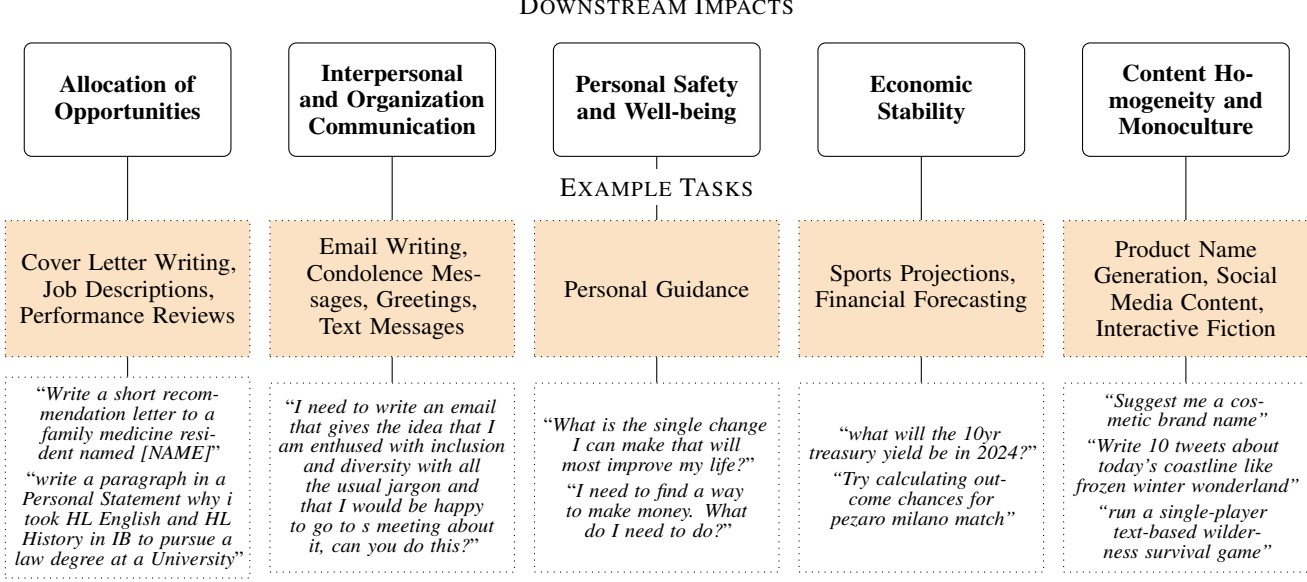

*Figure 2.* Creative composition tasks encompass use cases that need to be carefully evaluated to avoid harm in downstream applications. We highlight five areas where creative composition tasks that currently lack thorough evaluation may lead to undesirable consequences.

2024), an influx of homogeneous content may create undesirable incentives.

**Holistic Evaluation of Creative Outputs Should Also Consider Potential Harms** Since there are many potential ways in which the output of creative composition tasks may affect society, a holistic evaluation of creative output is necessary in order to assess the potential harms or unintended side effects of AI-generated content proliferation in our social systems. Additional dimensions of evaluation beyond novelty and value could include stereotypical associations, factuality, hallucinations (appropriate for the creative task), sycophancy, and distributional alignment. The call for holistic evaluations requires that both the quality and potential harms of creative output are examined. Certain tasks such as recommendation letter writing and character descriptions have been studied more frequently from the lens of representational harms than from the lens of quality of generation. We argue for the development of benchmarks that consider all of these dimensions.

## 5. New Paradigms for the Holistic Evaluation of Creative Composition Tasks

As users increasingly use language models for everyday tasks, interactions with ourselves and others become increasingly defined by the outputs of these models. While in the previous section, we highlighted some potential societal consequences of AI-augmented content, here we suggest

that a broadening of methodology is necessary to measure creativity in a scalable way. Metrics for group-level harms (e.g. stereotypes, discrimination) and societal harms (e.g. monoculture, misinformation) should be measured in conjunction with value and novelty for creative composition tasks. Since generalized multi-task creativity measurements have not yet been operationalized, there is an opportunity for holistic evaluations that consider societal impacts to become the standard practice for creative evaluation.

**Challenges of Generalized Creative Composition Benchmarks** Since creative composition tasks are missing from existing benchmarks, is the solution to simply include these tasks in existing benchmarks? We argue no — there are unique challenges to evaluating generation tasks with creative objectives. Moreover, developing creative benchmarks where evaluation scales to the growing number of new models is a challenging open scientific problem. A comprehensive evaluation of language models' creative composition capabilities would require new evaluation paradigms.

We highlight characteristics of existing benchmark metrics that do not extend to creative composition tasks. Generation tasks in existing benchmarks, such as translation and open-ended question answering, employ metrics that rely on several assumptions.

- **A reference answer(s)**: model generations are evaluated with respect to references based on similarity (e.g. F1 or BLEU scores). However, for creative tasks where

the goal is to produce novel and valuable artifacts, there is no correct reference answer. Always generating the same answer similar to a reference would be notably not novel. The value of a creative artifact should be evaluated based on task-dependent criteria rather than relative to a correct answer.

- **Single response evaluation**: For each translation or answer to a question, a conclusive score can be calculated independently. More responses from the same model or generations from different models are definitively not required. However, the novelty axis of creativity may require multiple candidate answers for comparison to properly judge the novelty of a specific answer.

- **Objectivity**: Although some disagreement may exist for the best translation or the best answer, the operationalization of benchmarks for these tasks assumes that the quality of these generations can be objectively evaluated. However, evaluating the value of different creative products can be highly subjective and may require multiple judges.

**Towards New Creative Composition Evaluation Paradigms** To better design metrics for the evaluation of creative composition capabilities, there are several axes of evaluation to consider that relate particularly closely to the societal impacts we have highlighted:

1. **Automatic evaluation with human oversight**: To evaluate every new model for creative abilities, automatic evaluation is likely the path forward in designing metrics across tasks for a comprehensive benchmark. However, LLM-as-a-judge exhibits many biases such as length (Hu et al., 2024; Dubois et al., 2024), position (Lu et al., 2021; Shi et al., 2024), and self-preference bias (Wataoka et al., 2024; Xu et al., 2024; Panickssery et al., 2024) that might hinder good evaluation. Moreover, when creating a creativity reward model, it is unclear whether human judgments can be consistently captured. Automatic creative evaluation may require better and pluralistic alignment (Sorensen et al., 2024), of LLM evaluators.

2. **Cohort-based evaluation**: Measuring the novelty of a single creative artifact can and should be enabled by metrics that compare a batch of other artifacts for the same creative tasks. The difference between a cliché and a novel solution may depend on whether all models produce the same creative artifact (Anderson et al., 2024; Bao et al., 2024; Padmakumar & He, 2023). For harms from monoculture, in particular, cohort-based novelty evaluation could be effective.

3. **Interactive evaluation**: The value of a creative product can expand beyond just a single interaction. When the goal is to co-brainstorm and better enable human creativity, proper evaluation of the creative capacities of models may be interactive. For example, in AI-human writing assistance or collaboration tasks: recent works propose interactive evaluation mechanisms for creative tasks (Lee et al., 2022; Mirowski et al., 2023; Chakrabarty et al., 2023).

## 6. Usage-Based Creative Composition Benchmarks: A Path Forward

Our position paper calls for usage-based benchmarks for creative composition tasks to better understand and intervene in the societal impacts of mass adaption of language models. Our recommendations are threefold: improved transparency of usage patterns, the development of broader usage-based evaluation, and further research on generated content in AI ecosystems.

**Transparency** To develop usage-based benchmarks, statistics and analysis of usage patterns are necessary. Open datasets collected by academic institutions enable our analysis. Tamkin et al. (2024) demonstrate that usage patterns can be studied while maintaining user privacy. Other LLM providers should follow suit to provide more information on understanding usage patterns. This information also further enables post-deployment monitoring of the foundation models more broadly, which may uncover new sociotechnical problems to be solved. A helpful analogy is social media research on lexical patterns of eating disorders that was enabled by access to Instagram APIs (Chancellor et al., 2016). Similarly, transparency in usage patterns can also enable better research on societal impacts of generated content.

**Evaluation** We need comprehensive benchmarks to measure the creative composition capabilities of the current suite of available models. Existing techniques from creativity evaluation (Lamb et al., 2018), such as rubric-based grading, interactive evaluation, and human feedback, can be combined with approaches for building standardized benchmarks such as automatic evaluation to create scalable evaluation of creative composition abilities of language and multimodal foundation models (Liang et al., 2022; Srivastava et al., 2022). Comprehensive benchmarks should include a representative set of tasks, as well as a combination of metrics for value, novelty, and societal impacts.

**AI Ecosystems Research** Beyond evaluation design, existing research on societal impacts in areas such as representative harms, monoculture, hallucinations, factuality, sycophancy, and agent behavior research can be strengthened by considering how users interact with systems day to day. While interaction patterns continue to evolve as both foundation models and user knowledge about foundation

models evolve. Examining how our current social systems will change as AI tools are increasingly used in creative composition and other tasks in an important line of work; particularly amid the development of AI agents. For example, understanding the strategic behavior of humans in augmenting application materials is important to ensure fair allocation of opportunities (Cohen et al., 2025).

## 7. Alternative Views

One alternative perspective to our position is that arena-style evaluation (e.g. Chatbot Arena (Chiang et al., 2024)) is sufficient for creative composition tasks. Since users also request creative composition tasks in the chat arena, win-rate scores within arena-style evaluations already measure creativity tasks. We take a data-driven objective to address this point. In our analysis, the distribution of creativity requests is long-tailed. For example, the count of a single user's fan-fiction prompts surpass all prompts that mention resumes. Under a pairwise-per-example score like Chatbot Arena, power users, and their niche use cases would dictate the rankings. Moreover, the creators of Chatbot Arena themselves highlight the development of task-specific arena rankings as future work. Their preliminary analysis reveals significant differences in win-rates for a top model (e.g. GPT4-0613) between topics (e.g. Python Programming 96.7% win-rate and movie recommendations 53.5% win-rate).

One may argue that creative fiction tasks (e.g., creative writing) should be separated from non-fiction tasks (e.g., cover letter writing) and societal impacts relate to the latter. Furthermore, all generations tasks can be considered creative tasks and it is hopeless to measure them all. We argue that, in practice, fiction vs. non-fiction tasks are difficult to delineate. Existing techniques from the Computational Creativity community include both fictional writing tasks and non-fiction problem solving tasks (Ismayilzada et al., 2024). This dichotomy overlooks the creativity required to write a compelling factual personal narrative or the productive creativity that is required for good business plans or proposals. Conversely, speculative tasks (e.g. "alternate Cold War timeline, where USSR survives collapse, 1975-1990") require imagining alternative realities based on historically accurate knowledge. Creating a flexible grouping of creative composition tasks using little-c creativity provides broader inclusion criteria for building a comprehensive benchmark. Furthermore, not all generation tasks require creativity. For example, requests to reformat or edit writing do not require novelty or present significant downstream harms to society.

## 8. Conclusion

Based on an in-depth study of large language model usage, we find that many usage patterns are not sufficiently cov-ered by existing benchmarks. Holistic evaluations for these tasks are necessary due to the downstream societal impacts of AI generated creative artifacts. Developing usage-based benchmarks that comprehensively evaluate the creative intelligence of foundation models is an important for both the AI creativity and societal impacts research communities.

## Acknowledgments

JHS is funded by the Simons Foundation Collaboration on the Theory of Algorithmic Fairness, the Sloan Foundation Grant 2020-13941, and the Simons Foundation investigators award 689988. CG is supported by Stanford Human Centered-AI. Thank you to Ed Chen, Liana Patel, Muneki Kawaguchi, Rishabh Ranjan, Yu Sun, and Mert Yuksekgonul for the helpful discussions around AI and creativity. Special thanks to Michelle Lam for suggesting different approaches for this problem and introducing the authors to a thematic analysis approach. Thanks to Rishi Bommasani for feedback on the draft of this paper.

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

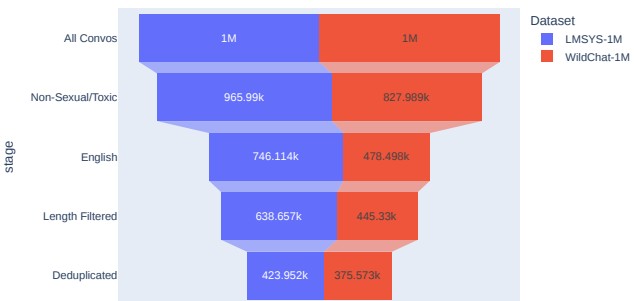

*Figure 3.* Prompt filtering pipeline: after applying moderation, language, length filtering an duplication, around 40% of total conversation were used for clustering and subsequent thematic analysis.

## A. Thematic Analysis: Creative Composition Usage Patterns

### A.1. Thematic Analysis

Thematic Analysis is a qualitative research method used to understand a dataset (Braun & Clarke, 2024). This technique is often used in the social sciences and also appears in human-computer interaction research (Adams et al., 2008). The focus of a thematic analysis is to identify patterns in a dataset from the bottom up. Our motivation for a deeper dive into prompt usage patterns was that previous analyzes present vague categories of usage (Table 1) but not enough specificity to develop usage-based benchmarks. We take the standard steps of thematic analysis including: generating initial codes, collating codes with supporting data, identifying themes, reviewing themes, and defining themes.

### A.2. Datasets

We examine two datasets WildChat-1M (Zhao et al., 2024) and LMSYS-Chat-1M (Zheng et al., 2023). LMSYS-Chat-1M contains 1 million conversations collected from April 2023 to August 2023 through three different chat interfaces on the LMSYS website[8] (single, anonymous side-by-side and self-selected side-by-side). Users gave consent for the release of their conversational data by accepting the terms of use. This data set contains mainly English conversations and conversations with the Vicuna model (Zhao et al., 2024). The most common use case, a third of the cases, reported is programming help; this included discussing software errors, questions about AI tools, Python coding assistance, and generating SQL queries. Another common use case, "*Language and Content Creation*" is reported to be around 30% of use cases and includes scenarios such as summarization, translation, and roleplaying of various characters. (Zheng et al., 2023) report that some clusters are generated in batches and submitted to their website.

The WildChat-1M dataset was collected through the Hugging Face Spaces platform from April 2023 to May 2024 (Zhao et al., 2024). This platform allowed users to chat with the GPT-3.5-Turbo or GPT-4 models. Users were presented with a data collection and sharing agreement before being given access to the chat interface. Just more than 50% of conversation were reported to be in English overall. The authors report that 61.9% of tasks where "assists/creative writing" and that the prompts they collected provide additional diversity compared to existing datasets. An additional analysis from (Longpre et al., 2024) estimate that creative composition queries for WildChat is the most common use case at almost 30% of queries.

**Data Filtering**   To understand the creative composition tasks in these datasets, we narrow the scope to analyzing the first user prompt. Since the publicly available version of WildChat already has toxic and sexual content filtered, we also apply this filter to the LMSYS prompts using the OpenAI moderation flags in the dataset. We filter for only English posts that are more than 5 words long. We also use exact match deduplication to remove repeated prompts, since we care only about unique use cases. Figure 3 shows the filtering process where we have 423K and 375K prompts remaining for LMSYS and WildChat respectively.

**Topic Clustering**   Following the same methodology as that used by (Zheng et al., 2023), we use sentence embeddings to cluster each data set into 50 clusters. The 100 clusters range from 1011 to 33463 prompts. We then apply a simple keyword

---

[8]https://chat.lmsys.org

search for each prompt to identify the proportion of creative prompts in each cluster. For each group with more than 30% prompts containing creative words, we sample 100 prompts and perform a thematic analysis of the creative composition use cases. We justify a simple key word approach by setting a low threshold for the percentage of creative prompts for inclusion in our analysis. This process produced 20 clusters and 154K prompts for LMSYS and 23 clusters and 135K prompts for WildChat.

**Final Datasets**    We make the prompts and clusters available here:

- LMSYS: https://huggingface.co/datasets/heyyjudes/lmsys-creative-labeled

- WildChat:       https://huggingface.co/datasets/heyyjudes/wildchat-creative-only-labeled

### A.3. Summary of Themes

For each cluster, we first manually code the different tasks that appear in the 100 sampled prompts and then ask a language model (Claude 3.5 Sonnet) to give a summary of the prompts. We take the union of manual and AI-identified tasks and assign general themes to the creative composition tasks. Although some clusters span multiple themes, most clusters focus on a single theme. Table 2 summarizes the 10 main themes identified; Each theme is accompanied by more specific tasks that were recorded, as well as representative examples from the sampled prompts.

#### A.3.1. IMAGINATIVE CREATIVITY

**Script and Story Writing**    A main theme that frequently appears and the most common category of tasks, in creative composition tasks, is script and story writing. Prompts across both datasets and many clusters ask for stories about specific fictional characters (fanfiction), new stories, and alternative realities. There was often a focus on generating detailed descriptions of character experiences and fantasy worlds, for example:

*Write a detailed story of A scuba diver undergoing a messy transformation into a mermaid.*

Some prompts also asked for descriptions of specific objects in fantasy worlds:

*In a game with merging mechanics for weapons, describe a weapon created by merging meatloaf with a spatula.*

The format of writing requests here also varies from scripts to interactive fiction games to short stories. Some prompts, for example, ask for a list of character ideas rather than full stories:

*Brainstorm unique and imaginary villains for a groundbreaking fantasy novel*

This may be multiple steps to story development where the first step is developing a character a user wants to read more about. In this category of fictional or creative writing tasks, users are likely seeking to entertain themselves by using LLM tools to generate new, but specific stories that they want to read.

**Poetry and Music Generation**    In both datasets, there was at least one group on the generation of poetry and music. These generation tasks range from writing poems to songs as well as generating playlists and other rhetorical devices such as proverbs. However, it is notable that these prompts are focused on very specific constraints rather than generating something to win the next poetry prize. For example:

*write a poem to purpose my girlfriend NAME_1, she is a marketing manager in a tech company.*

#### A.3.2. CREATIVE COMMUNICATION

**Advice and Interpersonal Communication**    In this category on composition tasks, contextual awareness may be more important than creativity. However, a significant number of advice-based prompts are related to brainstorming solutions for specific problems. For example:

*What is the single change I can make that will most improve my life?*

Although these questions are explicitly advice seeking, there is not one correct answer, and a user might be using the model for brainstorming and would thus benefit from a diverse set of useful suggestions. Other prompts on this theme include writing text messages, greetings, and other personal messages.

**Title, Name, and Phrase Generation**   Many prompts explicitly asked for search-engine-optimized (SEO) product descriptions and hashtags. Another common use case was asking for names, titles, and slogans for different products and characters. Many of these prompts explicitly asked for multiple suggestions. For example:

> *brainstorm a creative list of username for my personal branding account. it must contain word 'abito' in it somewhere.*

In asking for a list of possible suggestions, the user benefits from both high-quality suggestions and diversity in suggestions.

**Professional Writing**   Another frequent category of prompts is professional writing tasks. Users ask for help with writing emails, reviews, feedback, articles, blogs, job descriptions, cover letters, and resumes. They ask for help both in the drafting process and in the editing process. Some of these tasks are very high stakes, for example, personal statements:

> *write a paragraph in a Personal Statement why i took HL English and HL History in IB to pursue a law degree at a University*

In these tasks where the generated material may be evaluated by humans or other applicant screening systems, infusing creativity into the writing would help an application stand out. Since both hiring and admissions committees may see thousands of resumes and personal statements, generating a unique and high-quality response would be helpful for the user. Other prompts ask to elicit sincere communication, for example:

> *I need to write an email that gives the idea that I am enthused with inclusion and diversity with all the usual jargon and that I would be happy to go to s meeting about it, can you do this?*

In these types of professional writing, diverse outputs to choose from may better help convey authenticity.

### A.3.3. PRODUCTIVE CREATIVITY

**Business Development**   A less common but also business-oriented theme is marketing and business development ideas and proposals. These prompts relate to different segments of the business development process from brainstorming ad campaigns to corporate strategy. Business schools have long touted the importance of creativity and design thinking in the business world (Boyles, 2022); the prompts in this category illustrate that users might be turning to language models for this creativity. For example, some users ask for a full business plan based on a light description:

> *can i start my own business as a real estate agent in NAME_1,can you help me draft a detailed robust business plan*

while other users want a list of potential directions to pursue:

> *Give me a list of ideas for a startup, created by a computer science student with zero to little inversion.*

These tasks fit under the notion of business creativity and innovation. Ideas that are both unique (compared to existing approaches) and strategic are helpful for these users.

### A.3.4. CREATIVE INQUIRY

The four themes under creative inquiry encompass argument development and critical thinking. Application domains include sports, education, and scientific development. The goals in these creative composition tasks is to create original or novel developments.

**Argument Composition**   Several prompts explicitly ask for reasoning and arguments for specific topics. Some topics may require access to facts but other arguments are altogether hypothetical. There may not be right or wrong answers to these questions but rather different perspectives. For example:

*is free trade a good or bad thing*

Diverse but thoughtful arguments in the responses to these questions would help the user better engage with the topic. When these topics are applied to domains such as debate, creativity is valued for developing novel, yet convincing arguments.

**Academic Writing**   Another theme is where users explicitly ask for help with educational tasks. These prompts include various stages of writing help from generating essay outlines to drafting and revising essays. Also in this category are prompts asking to learn or teach certain concepts, for example:

*I want you to act as a data science instructor. Explain what neural network is to an undergraduate.*

Many educators have highlighted the role of creativity in effective teaching (Jeffrey* & Craft, 2004; Beghetto, 2017). A user prompting the language model may want effective and novel ideas for explaining or illustrating concepts.

**Scientific Development**   Using AI for ideation has recently been studied for various scientific pursuits from material discovery (Toner-Rodgers, 2024) to prompt engineering (Si et al., 2024). This theme also appeared in our analysis, for example:

*give me interesting Human-Computer interaction project can be done using intel realsense 3D camera*

Users gave specific topics and constraints in hopes of generating ideas for research. Another common mode under this theme is the generation of prompts for tasks. For example prompts start with:

*You are a helpful Stable Diffusion prompt generator.*

There may be one answer that is better than others but a group of diverse prompts may ultimately help generate a better set of outputs.

**Forecasting**   A less common but distinct theme is the usage of LLMs for projections or forecasting. There were many requests for sports outcomes and scripts of upcoming matches as well as financial market predictions. Since the future in inherently unpredictable, diverse projections may improve coverage of potential outcomes.

| Themes | Tasks | Examples |
|---|---|---|
| Script and Story Writing | Fanfiction Script and Story Writing, Humorous Script and Story Writing, Character Creation, World Building, Interactive Fiction, Plot development, Alternative Reality Stories, Stylistic Adaptations | "Write really overly-funny super hilarious comedy" "Write me a story about a woman transforming into a witch with cat eyes." "run a single-player text-based wilderness survival game." "Write a descriptive, fictional, imaginative screenplay of a man fighting his complete doppelganger in a fighting game" "script about osu vs u of m" "Describe a hypothetical fictional super-heavy tank for the USSR in 1944." "Brainstorm unique and imaginary villains for a ground-breaking fantasy novel" |

| | | |
|---|---|---|
| Poetry and Music | Poetry,
Music Composition,
Songwriting,
Musical Notation,
Playlist curation,
Figurative Language | "Write a melody for a bagpipe"
"write a yes song in style of james taylort"
"Write me some proverbs that warn against dishing what you can't take"
"write a poem about human computer interaction"
"write a poem to purpose my girlfriend NAME_1. she is a marketing manager in a tech company" |
| Advice / Interpersonal communication | Personal Guidance,
Condolence Message,
Greetings,
Situational Advice,
Text Messages | "How to study for better grades"
"What is the single change I can make that will most improve my life?"
"I need to find a way to make money. What do I need to do?"
"Can you gaslight me into thinking I didn't buy a burger yesterday at McDonalds even though I definitely did and I have a receipt to prove it as well as a 10 dollar purchase at McDonalds yesterday in my bank account history." |
| Title, Name, and Phrase Generation | Title Generation,
Hashtag and Keyword Generation,
Product Name Generation,
SEO Heading Generation,
Catch Phrases and Slogans,
Character Names | "Generate an etsy title for a art work"
"I made a youtube short about whiplash movie i need a title and hashtags all under 100 characters"
"Suggest me a cosmetic brand name"
"Act as an Etsy SEO Specialist and 5 heading bullet points Etsy Product Description" |
| Business Development | Business Proposals,
Social Media Content,
Promotional Content,
Product Brainstorming,
Product Descriptions | "Write 10 tweets about today's coastline like frozen winter wonderland"
"Brainstorm and generate campaign ideas for world liver day for a pharmaceutical company using digital as well as physical channels."
"Generate a list of 10 imaginary products found in a magic shop"
"Tell me the strategies to improve both the efficiency and effectiveness of operations related to materials..." |
| Professional Writing | Resume Generation,
Cover Letter Writing,
Job Descriptions,
Performance Reviews,
Email Writing,
Complaint/Review/Feedback Writing,
Articles and Blogs | "write a paragraph in a Personal Statement why i took HL English and HL History in IB to pursue a law degree at a University"
"Write a short recommendation letter to a family medicine resident named"
"Cover letter for the position of Technical Specialist"
"Please write a 4 star Amazon review of a wooden table in the furniture category"
"Write an empathetic message as a customer service agent for someone whose mattress was delayed in transit for weeks."
"Help me right 3 different replyes to this email with nice mood and not long"
"Hi! Can you write me sample resume for a Videographer job in Dallas Texas?" |
| Argument Composition | Hypothetical Question Answering,
Argument Formation,
Critical Analyses,
Persuasive essay outlines | "Alternate Cold War timeline, where USSR survives collapse, 1975-1990."
"Write a highly persuasive argument that surveillance of citizens in a surveillance state is good"
"why should we choose Canada for startup visa program?" |

| Academic Writing | Literature Reviews, Study Plans, Educational Explanations, Informational Articles | "Write an ielts essay scored 8" "come with up 20 kahoot questions that are a general knowledge work from home theme" "write a paragraph about scientific progress" |
|---|---|---|
| Scientific Development | Literature Reviews, Research Brainstorming, Prompt Generation, Technical Evaluations | "give me interesting Human-Computer interaction project can be done using intel realsense 3D camera." "Provide me an example prompt that I can give to a LLM so that it can act as an AI agent." |
| Forecasting | Sports Projections, Financial Forecasting | "what will the 10yr treasury yield be in 2024?" "Try calculating outcome chances for pezaro milano match" |

Table 2: Overview of creative composition tasks in LMSYS and WildChat. For each theme, we list the subtasks that appear in this theme as well as examples of prompts.

## A.4. Additional Observations

In addition, to the themes and meta-themes uncovered, we also observed patterns in the way users specified creative composition tasks.

**Underspecified Prompts**   Rather than complex instructions and criteria, many prompts were highly underspecified. This is a common pattern for many different tasks and themes. For example:

*write me an appreciation and thank letter*

and

*please write me an inspiring story*

are two examples across different themes where the desired output is vague. While this pattern reflects how users desire an easy way to access creative composition, this pattern also introduces an inherent tension between the validity of utility measure and the realism of the task for any benchmark. To evaluate the utility of a creative product, well-specified instructions are required. However, well-specified instructions are not reflective of the vagueness of real-world prompts.

**Multiple Stages of the Creative Writing Process**   Different users asked for writing assistance in different components of the writing process. For example, some tasks ranged from ideation:

*Please design a DLC storyline for Pokémon: Sun/Moon based on the Call of Cthulhu.*

to drafting:

*write a paragraph about scientific progress*

to editing:

*can you make this email nicer*

Although all these stages appear in user prompts for creative composition tasks, the ideation and drafting stages likely allow a larger creative composition space. Moreover, more existing creativity measurements focus on the drafting stage (Lu et al., 2024; Chakrabarty et al., 2024) than the ideation stage (Si et al., 2024).

| Creative Terms | Non-Creative Terms |
| --- | --- |
| imagine, envision, innovate, inspire, story, poem, novel, write, draw, paint, compose, art, brainstorm, idea, suggestion, argument, creative, fake, act, new, novel | code, python, problem, solution, c#, app, api, client, serve, prompt, prompts, chatgpt, openai, lua, java |

*Table 3.* List of Creative and Non-Creative Terms Used for Filtering.

**Power Users** Our analysis found that certain users, or groups of users, dominated entire clusters. This finding is consistent with (Zheng et al., 2023). We categorized power users into three main categories: (1) LLM testing (LMSYS), fanfiction (LMSYS and WildChat), and customized stories (WildChat). For example, of the creative clusters we analyzed, we discovered the following large clusters of prompts:

- 1671 prompts about Jane, the American TV Series

- 10311 prompts about the anime Doki Doki Squad

- 3643 prompts for customized stories about David and sexual hypnosis scenarios

- 4441 prompts for a specifically formatted fact-checking task

- 4058 prompts on Freedom Planet Fanfiction and Crossovers

- 6986 prompts on writing the introduction of a chemical company

- 1996 prompt testing logical reasoning of a model called "SmartGPT"

- 14006 prompts about writing an article about a specific chemical in the chemical industry

These clusters persisted even after our de-duplication since each prompt is actually unique and even rewritten in various ways for the fan-fiction tasks. Prompts for sexual content persisted even after filtering for adult and toxic content; both datasets contained several creative task clusters (around 15 - 25k prompts) of prompts for sexual story writing. This finding is relevant not only for studying creative composition but also for all users of these two datasets. The high concentration of these prompts may lead to desirable downstream outcomes (e.g., the objectification of certain populations).

**Creative Word List** We used the following list of words to identify clusters that may contain creative tasks. Table 3 shows the creative and non-creative terms we used. We labeled a prompt creative if it contained at least one word from the creative list and did not contain any words from the non-creative list.

