# OpenReview forum: "Position: Societal Impacts Research Requires Benchmarks for Creative Composition Tasks"
_ICML.cc/2025/Position_Paper_Track — ICML 2025 Position Paper Track poster_

### Official Review · Reviewer_nwGp · 2025-03-13

**Significance:** 3
**Argument Clarity:** 3
**Rating:** 4
**Confidence:** 4

**Questions:**

In addition to some questions raised in the "weaknesses" above (continuing numbering from there):

4. Is the analysis of the WildChat-1M and LMSYS-Chat-1M datasets novel?

5. I am confused by the terminology of "creative composition tasks". It seems this is from the literature, but to me, the paper is simply discussing "everyday tasks" or "usage-based tasks", and calling some of these creative feels misleading.

**Discussion Potential:**

2

**Paper Summary:**

This paper argues for AI benchmarks that are more grounded in real-world usage patterns. Their argument includes discussion of:
1. Existing benchmarks do not cover common usage patterns
2. Why areas neglected by existing benchmarks are important
3. Suggestions for improving evaluation paradigms

**Position:**

Yes

**Position In Title:**

Yes

**Related Work:**

3

**Strengths And Weaknesses:**

The stated position is well-supported with reasoning and evidence. The topic is highly relevant and important to the ICML community, although the main contribution is highlighting a position that is under-discussed (not one that many should disagree with!). Overall, the paper has many strengths with only some minor weaknesses in my view.

**Strengths**:
- There have been limited empirical analyses of real-world LLM usage patterns, and the analysis presented by the authors appears to be novel. The categorization of everyday tasks in Figure 1 is a strong contribution, given it is grounded in real-world evidence.
- The subsection "creative composition use cases impact many aspects of society" is highly relevant and well-written. The key examples highlighted by the authors are not addressed by current evaluations and benchmarks.

**Weaknesses**:
1. The title is a little clunky (and originally made me wary of the paper). While it accurately represents the authors position, I wonder if something like "Generative AI Requires Usage-Based Benchmark Development To Understand Societal Impact" would be more clear.
2. Figure 1 is useful, but I am wondering if it could be condensed. Perhaps the citations to existing benchmarks could be moved to the appendix.
3. Sections 5 and 6 could perhaps be more specific with actionable insights. For example, the "towards new creative composition evaluation paradigms" feels more retrospective than forward-looking. How should creative rewards be captured instead of using LLM-as-a-judge? Also, interactive evaluation seems to advocate for evaluating users (e.g. through HCI surveys), but this is not clear. Furthermore, LLM companies are unlikely to release usage data, how could this be addressed?

**Support:**

3

---

> ### Author Rebuttal · Authors · 2025-03-27
>
> We thank the reviewer for the review and the comments. The title suggested, “Generative AI Requires Usage-Based Benchmark Development To Understand Societal Impact,” is very nice and captures what we are trying to say. Makes sense to adapt it for our paper! We will focus on responding to the reviewer's questions in this rebuttal:
>
> > Sections 5 and 6 could perhaps be more specific with actionable insights. For example, the "towards new creative composition evaluation paradigms" feels more retrospective than forward-looking. How should creative rewards be captured instead of using LLM-as-a-judge? Also, interactive evaluation seems to advocate for evaluating users (e.g. through HCI surveys), but this is not clear. Furthermore, LLM companies are unlikely to release usage data, how could this be addressed?
>
> While we point out the limitations of LLMs-as-a-judge. Our position is not to rule out automatic evaluation but to recommend carefully designed evaluation. For example, more robust automatic evaluation may involve aggregating the preferences of multiple judges (e.g. pluralistic alignment methods) or incorporating human preferences for specific tasks (e.g. training reward models for specific creative tasks). While our goal is to give general suggestions rather than prescriptive methods, we will add more examples to Sections 5 and 6 for further specificity.
>
> In terms of usage data release, this is a great point! Clio [2] has demonstrated that “privately” releasing data on usage patterns is possible. Mechanisms for encouraging further data release might be in the form of policy requirements (EU AI Act Article 72) or through non-industry parties hosting platforms (e.g. LMSYS). We will include a discussion of this in our paper.
>
> > Is the analysis of the WildChat-1M and LMSYS-Chat-1M datasets novel?
>
> To the best of our knowledge, a thematic analysis [1] of LLM prompting patterns has not been done before. There are existing high-level statistics included in the dataset release papers of WildChat-1M and LMSYS-Chat-1M. But these statistics are not sufficient to understand the detailed tasks for which benchmarks are required. The closest work to ours is concurrent work (released shortly before the ICML deadline) from Anthropic: Clio [2]. They use an internal proprietary tool to analyze these public datasets into coarse categories. Thus, we do believe our thematic analysis is unique and one of the first detailed thematic analyses of these datasets (both datasets were only released in the last 18 months).
>
> > I am confused by the terminology of "creative composition tasks". It seems this is from the literature, but to me, the paper is simply discussing "everyday tasks" or "usage-based tasks", and calling some of these creative feels misleading.
>
> Typically, “creative” tasks described in AI refer to Big-C or Pro-C creativity, where AI models are developed and measured for their ability to perform at the level of artists or professionals in creativity. We call for a focus on little-c (also known as ordinary creativity tasks). Hence, we introduce the term “creative composition tasks” to describe this category of everyday tasks that have significant societal impact. Indeed, these tasks are not “creative” in the sense of Big-C or Pro-C but it is nevertheless considered creative in the little-c sense. We highlight this in Section 2.1, but we are happy to further clarify it throughout our work.
>
> ## References
> [1] Guest, G., MacQueen, K. M., and Namey, E. E. Introduction to applied thematic analysis. Applied thematic analysis, 3(20):1–21, 2012.
>
> [2] Tamkin, A., McCain, M., Handa, K., Durmus, E., Lovitt, L., Rathi, A., ... & Ganguli, D. (2024). Clio: Privacy-Preserving Insights into Real-World AI Use. arXiv preprint arXiv:2412.13678.

---

### Official Review · Reviewer_6ZNZ · 2025-03-13

**Significance:** 3
**Argument Clarity:** 3
**Rating:** 3
**Confidence:** 2

**Questions:**

No

**Discussion Potential:**

3

**Paper Summary:**

This position paper argues that existing evaluation benchmarks for large language models (LLMs) fail to capture real-world creative usage patterns, which has significant societal implications. The authors conduct a thematic analysis of two million user prompts from public LLM interaction datasets, identifying "creative composition tasks" as a major category of LLM usage. They claim that current benchmarks inadequately measure these tasks, which include personal and professional writing, brainstorming, and content creation. The paper contends that neglecting these use cases can lead to unintended societal harms such as content homogenization, biases in professional communication, and misinformation. It advocates for developing benchmarks that align with real-world LLM applications, incorporating multi-objective evaluation paradigms that assess creativity, societal impact, and ethical concerns.


### Update After Rebuttal
After reading the response, I am still positive about this paper.

**Position:**

Yes

**Position In Title:**

Yes

**Related Work:**

3

**Strengths And Weaknesses:**

Strengths

1. The paper correctly identifies a gap in existing LLM benchmarking, highlighting how current benchmarks fail to represent real-world creative applications.
2. The large-scale analysis of user prompts (2 million samples) provides concrete evidence supporting the claim that creative composition tasks are a dominant LLM use case.
3. The discussion on potential harms (e.g., bias in professional writing, misinformation in brainstorming tasks) is a meaningful contribution to AI ethics and impact research.
4. The argument for multi-dimensional evaluation (including novelty, factuality, and societal risks) is valid and could improve the robustness of AI evaluations.

Weaknesses

1. While the paper argues for new benchmarks, it does not offer a well-defined framework or methodology for how these should be designed and implemented. The paper remains at a high level without clear operational steps.
2. The definition of creative tasks is broad, encompassing everything from fiction writing to resume drafting. This generalization dilutes the argument, as different tasks have distinct evaluation needs.
3. The paper discusses societal impacts such as bias, misinformation, and content homogenization but provides little direct evidence on how these problems manifest in existing AI systems. The discussion remains speculative rather than empirical.
4. While the paper briefly addresses opposing perspectives (e.g., existing arena-style evaluations are sufficient), it does not engage deeply with counterarguments or propose a comparative analysis of current evaluation methods.
5. The call for "holistic evaluation paradigms" lacks a feasibility analysis—how scalable or practical would these new evaluation methods be? The paper does not address the challenges of human-centered creativity evaluation in AI.

**Support:**

3

---

> ### Author Rebuttal · Authors · 2025-03-27
>
> We appreciate the reviewer's thoughtful evaluation of our position paper. We would like to provide some clarifications to the weaknesses highlighted.
>
> > (1) While the paper argues for new benchmarks, it does not offer a well-defined framework or methodology for how these should be designed and implemented. The paper remains at a high level without clear operational steps.
> > (3) The paper discusses societal impacts such as bias, misinformation, and content homogenization but provides little direct evidence on how these problems manifest in existing AI systems. The discussion remains speculative rather than empirical.
>
> We recognize that as a position paper, our contribution is primarily conceptual, offering a thematic analysis of creative tasks using real-world datasets of 2 million prompts and highlighting connections to consequential societal impacts. Thus, our focus is not on providing empirical results or detailed implementation guidelines. We reference many works that present empirical approaches to measuring different aspects of societal impacts that can be applied to creative composition tasks as well. We aim to stimulate discussion about evaluation approaches for these increasingly important use cases. If the reviewer has specific questions about our thematic analysis or proposed directions, we would be happy to address them.
>
> >(2) The definition of creative tasks is broad, encompassing everything from fiction writing to resume drafting. This generalization dilutes the argument, as different tasks have distinct evaluation needs.
>
>
> Regarding the broad definition of creative tasks: We address this in "Alternative View 2," where we explain that our coverage is based on existing review work in computational creativity and the interconnectedness of these tasks.
>
> >(4)While the paper briefly addresses opposing perspectives (e.g., existing arena-style evaluations are sufficient), it does not engage deeply with counterarguments or propose a comparative analysis of current evaluation methods.
>
> Additionally, we do address current evaluation methods in Section 3, where we argue that current benchmarks are not representative of creative usage patterns.
>
> >(5) The call for "holistic evaluation paradigms" lacks a feasibility analysis—how scalable or practical would these new evaluation methods be? The paper does not address the challenges of human-centered creativity evaluation in AI.
>
> The reviewer makes a great point about better addressing the limitations of our analysis. We do not include a feasibility analysis because it would heavily depend on the task measured. For example, designing benchmarks for resumes and personal statements would need privacy considerations. These considerations are important but beyond the scope of this position paper.
>
> We are happy to answer any further questions!

---

### Official Review · Reviewer_PxLe · 2025-03-18

**Significance:** 4
**Argument Clarity:** 4
**Rating:** 3
**Confidence:** 3

**Questions:**

Major questions:
- What is a usage-based benchmark exactly?
- Is to goal to assess if LLMs can do little-c creativity, or is it about the human seeking assistance with their little-c creativity?

Minor questions:
- How do you envision it limits homogenous content?
- *L267-269*. The authors argue that figure 1 shows a gap between existing benchmarks and detailed usage patterns, but I don't see it. What do you mean?

**Discussion Potential:**

3

**Paper Summary:**

This position paper argues that to investigate the societal impact of generative AI, we need creative task benchmarks.
The authors focus specifically on what they call, a creative composition task, which focuses on the everyday creativity involved in composing artefacts, such as cover letters and personal statements.
They argue for their position in four steps. Firstly, they argue that existing benchmarks do not suffice in evaluating creative usage patterns. Secondly, they show areas neglected by the benchmarks.
Thirdly, they show why existing methods do not extend to creative composition tasks.
Lastly, they argue that more transparency of usage patterns of foundation models will help with the development of the kind of benchmarks envisioned in this paper.
The authors also present two alternative views: 1) current benchmarks are good enough, and 2) creative tasks have no societal impacts.

**Position:**

Yes

**Position In Title:**

Yes

**Related Work:**

3

**Strengths And Weaknesses:**

### Strengths

The paper is very well-written and argues for a very important aspect, the creativity involved with composing text, that I agree is currently absent in the evaluation of LLMs.
Section 3, in particular, nicely argues, in my opinion, how current benchmarks do not suffice, while also highlighting that usage of LLMS is largely focused on stimulating (little-c) creativity, hence motivating the position.
Overall, the ``why'' is appropriately addressed, and the position is well-grounded in the literature with plenty of examples, although the creativity perspective specifically, could be deeper.


### Weaknesses

The main weakness of this paper is that I'm not sure what the authors mean by a "usage-based benchmark", it is not clearly defined. How do the authors envision this would work for their creative composition task?
Since this is a key part of their position, this is worrying. While sections 5 and 6 describe characteristics and criteria for the evaluation of such tasks, I miss a practical example of what a particular instance in a task looks like.

While the authors describe in section 5 ideas towards evaluation, I find the discussion limited.
The particular challenge with something like little-c creativity is that its personal and context-specific. Benchmarks, however, are taking, almost by definition, a general approach, and sort of smooth out precisely what little-c creativity is about.

I imagine, that for novelty in particular it would require benchmarks that consist of several counterexamples and that the goal is to generate something different by some distance (using some metric). Or perhaps, very similar (but not identical) prompts with very different answers (reducing homogeneity). Overall, I believe ideas about the "how" question with respect to the actual benchmark design could have received more attention and depth.

Another key point that could be stronger from this paper is the evaluation goal. Is it the LLM exhibiting little-c creativity, or the human seeking assistance with their little-c creativity? Almost all examples of tasks in this paper address the latter, but the authors don't take a clear stance. In the latter case, I would argue this changes what kind of answer is required, as it would assess if the output is novel or valuable to the user, as opposed to the LLM.

I'm a little confused about dedicating a full page to figure 1, while it gives a good overview of creative task benchmarks and those that are missing. I found myself more interested in seeing some examples of usage patterns.

### Other Remarks
- "task" in the abstract is not italics.
- I think it would be good to cite the papers that originated the standard definition of creativity of novelty and value.
    - Stein, M. I. (1953). Creativity and Culture. The Journal of Psychology, 36(2), 311–322. https://doi.org/10.1080/00223980.1953.9712897
    - Barron, F. (1955). The disposition toward originality. The Journal of Abnormal and Social Psychology, 51(3), 478–485. https://doi.org/10.1037/h0048073
    - Runco, M. A., & Jaeger, G. J. (2012). The Standard Definition of Creativity. Creativity Research Journal, 24(1), 92–96. https://doi.org/10.1080/10400419.2012.650092
- *L106 C1* A useful extension of the 4Ps for computation systems and creativity:
    - Jordanous, A. (2016). Four PPPPerspectives on computational creativity in theory and in practice. Connection Science, 28(2), 194–216. https://doi.org/10.1080/09540091.2016.1151860
- *L414 C1* More appropriate would be "Existing techniques from the Computational Creativity community". This community has developed extensive frameworks and methods to approach the complex subject of creativity evaluation of computational creativity.

**Support:**

3

---

> ### Author Rebuttal · Authors · 2025-03-27
>
> We would like to thank the reviewer for appreciating our contributions and providing suggestions for strengthening our paper. We will definitely add the additional works to our paper as well as expand on specific examples and guidance for designing evaluations. To adhere to the response period guidelines, we will focus on the reviewer's questions in our response:
> 1. **Usage-based benchmark**: A usage-based benchmark evaluates LLM performance on tasks derived from observed real-world user interactions. In contrast to synthetic or template-based benchmarks, these better reflect how LLMs are used in practice. For example, based on our thematic analysis, benchmarks around personal statement and advice are “usage-based” since we observed usage patterns for these cases. In existing literature, terms such as “natural” [1, 2] data or “realistic” [3, 4] appear to describe benchmarks or datasets created based on real-world usage. However, these terms tend to mean many different things. We'll revise our terminology to 'real-world benchmarks' throughout the paper for clarity and provide specific examples of implementation.
> 2. **Creativity Assessment Goals**: In these LLM transcript datasets, usage is primarily centered around humans seeking LLM assistance with their little-c creative tasks. This distinction is indeed important for benchmark design. We will clarify that better in our paper. Value to the user is an important dimension to measure, but it might be subjective and difficult. Thus, we imagine proper evaluations to be similar to approaches from pluralistic alignment, where different personas or cultures have different preferences that are measured. We will clarify this further in our work.
> 3. **Homogeneous Content**: We envision creativity-based benchmarks to require cohort-based evaluation in order to measure output diversity between models. Certain models may generate very similar things to each other, while other models might generate answers different from the pack. This is just one axis (novelty) on the novelty-value axis that might be used to measure a creative product. We believe this is undermeasured in platforms like LMSYS ChatArena with feature side by side comparisons. For example, imagine a simple update where three models are shown beside each other, then the more novel model might be favored in creative composition tasks the two other models generate the same content. This might produce a different winner than a head-to-head comparison of only 2 models.
> 4. **Figure 1**: Thank you for highlighting this confusion. In Figure 1, the transparent boxes represent task categories where benchmarks already exist, while yellow boxes highlight task categories where our analysis identified significant user activity but no corresponding benchmarks. We'll revise both the figure and accompanying text to make this distinction more obvious and quantify the specific gaps more precisely.
>
> We thank the reviewer for all the helpful feedback. We aimed not to be too prescriptive with the guidance for benchmark development, but we will add additional details to ensure our proposals for the types of benchmarks we’d like to see are clearly presented. Please let us know if we can answer any additional questions.
>
> ## References:
> [1] Kwiatkowski, T., Palomaki, J., Redfield, O., Collins, M., Parikh, A., Alberti, C., ... & Petrov, S. (2019). Natural questions: a benchmark for question answering research. Transactions of the Association for Computational Linguistics, 7, 453-466.
>
> [2] Meyer, S., & Elsweiler, D. (2022, June). Glohbcd: A naturalistic german dataset for language of health behaviour change on online support forums. In Proceedings of the Thirteenth Language Resources and Evaluation Conference (pp. 2226-2235).
>
> [3] Su, H., Yen, H., Xia, M., Shi, W., Muennighoff, N., Wang, H. Y., ... & Yu, T. (2024). Bright: A realistic and challenging benchmark for reasoning-intensive retrieval. arXiv preprint arXiv:2407.12883.
>
> [4] Gu, G., Zhao, Y., Ning, R., Zheng, Y., & Cohan, A. (2024, November). TAIL: A Toolkit for Automatic and Realistic Long-Context Large Language Model Evaluation. In Proceedings of the 2024 Conference on Empirical Methods in Natural Language Processing: System Demonstrations (pp. 198-208).

---

### Decision · Program_Chairs · 2025-04-30

**Decision:**

Accept (poster)

**Comment:**

The paper presents a very interesting idea. Perhaps not only the societal impacts but even the basic quality of AI should be judged using creative tasks. Appropriate benchmarks are therefore required. The reviewers were positive, and there is no reason to reject this paper.